# Mapping Vegetation Types by Different Fully Convolutional Neural Network Structures with Inadequate Training Labels in Complex Landscape Urban Areas

**Shudan Chen [1,2,3], Meng Zhang [1,2,3],\*, and Fan Lei [4],\***

1   Research Center of Forestry Remote Sensing & Information Engineering, Central South University of Forestry and Technology, Changsha 410004, China; 20211100019@csuft.edu.cn
2   Key Laboratory of State Forestry Administration on Forest Resources Management and Monitoring in Southern Area, Changsha 410004, China
3   Key Laboratory of Forestry Remote Sensing Based Big Data & Ecological Security for Hunan Province, Changsha 410004, China
4   Hunan Second Surveying and Mapping Institute, Changsha 410000, China
\*   Correspondence: mengzhang@csuft.edu.cn (M.Z.); cnclcn@foxmail.com (F.L.)

**Abstract:** Highly accurate urban vegetation extraction is important to supporting ecological and management planning in urban areas. However, achieving high-precision classification of urban vegetation is challenging due to dramatic land changes in cities, the complexity of land cover, and hill shading. Although convolutional neural networks (CNNs) have unique advantages in remote sensing image classification, they require a large amount of training sample data, making it difficult to adequately train the network to improve classification accuracy. Therefore, this paper proposed an urban vegetation classification method by combining the advantages of transfer learning, deep learning, and ensemble learning. First, three UNet++ networks (UNet++, VGG16-UNet++, and ResNet50-UNet++) were pre-trained using the open sample set of urban land use/land cover (LULC), and the deep features of Sentinel-2 images were extracted using the pre-trained three UNet++ networks. Subsequently, the optimal deep feature set was then selected by Relief-F and input into the Stacking algorithm for urban vegetation classification. The results showed that deeper features extracted by UNet++ networks were able to easily distinguish between different vegetation types compared to Sentinel-2 spectral features. The overall classification accuracy (OA) of UNet++ networks and the Stacking algorithm (UNS) was 92.74%, with a Kappa coefficient of 0.8905. The classification results of UNet++ networks and the Stacking algorithm improved by 2.34%, 1.8%, 2.29%, and 10.74% in OA compared to a single neural network (UNet++, VGG16-UNet++, and ResNet50-UNet++) and the Stacking algorithm, respectively. Furthermore, a comparative analysis of the method with common vegetation classification algorithms (RF, U-Net, and DeepLab V3+) indicated that the results of UNS were 11.31%, 9.38%, and 3.05% better in terms of OA, respectively. Generally, the method developed in this paper could accurately obtain urban vegetation information and provide a reference for research on urban vegetation classification.

**Keywords:** urban vegetation; classification; UNet++ networks; Stacking; Sentinel-2

## 1. Introduction

Vegetation is an important component of the urban environment, improving urban air quality and reducing the heat island effect [1], thereby reflecting the state of the regional ecological environment. However, the dramatic change in urban land use as cities expand into the periphery leads to changes in regional ecosystems and environmental problems [2]. Therefore, accurate urban vegetation classification studies are important for urban ecosystem and sustainable urban development [3,4].

Early urban vegetation extraction relied on field surveys, which were demanding and lengthy enough to support vegetation classification in areas with complex topography and

diverse land cover types. With the development of remote sensing technology, it is now possible to monitor complex environments with the advantages of rapid data collection and cost savings [5]. Remote sensing data sources started with optical images at low-to-medium spatial resolution, such as national oceanic and atmospheric administration (NOAA)/advanced very high-resolution radiometer (VHRR), medium-resolution imaging spectroradiometer (MODIS), Landsat, and SPOT [6,7]. In recent years, high-resolution sensors (Sentinel-2, IKONOS, QuickBird, and GF series, etc.) have met the need for high-precision extraction of vegetation information as the availability and spatial resolution of historical remote sensing data has improved [8,9]. For instance, Sentinel-2 has been widely used in vegetation extraction and land use classification studies due to its easy accessibility, wide spatiotemporal resolution, and rich spectral bands [10,11]. However, urban vegetation classification was susceptible to spectral clutter and shading (variations in mountainous terrain) when using satellites such as Sentinel-2 to obtain spectral information [12], which prevented accurate results from being obtained.

Approaches to the early vegetation classification were mostly based on mathematical statistical principles, such as maximum likelihood in supervised classification, minimum distance classification, and iterative self-organizing data analysis in unsupervised classification. Nevertheless, they required random uniform and normal samples; thus, various machine learning (ML) algorithms, such as support vector machines (SVM), decision trees, and random forests (RF), have been proposed and employed in vegetation classification studies [13–15]. Normally, single ML algorithms have some inherent limitations, with potential instability across data and scenarios [16]. To solve this problem, a combination of multiple weak classifiers using the idea of ensemble learning (bagging, boosting, and Stacking) was able to obtain a better supervised result. Bagging and boosting used the simple idea of voting and averaging, while Stacking used the idea of weighting when combining the results of different classifiers, facilitating the extraction of vegetation information in highly heterogeneous areas [17,18].

Compared to ML, convolutional neural networks (CNNs) are capable of analyzing the information of adjacent pixels and better extracting image features. This has resulted in better results in image classification researches [19,20], and convolutional neural networks such as fully convolutional networks (FCNs), e.g., DeepLab V3+, are consequently becoming increasingly popular in LULC and vegetation classification [21,22]. FCNs have replaced the last layer in CNNs with a deconvolution [23], which has retained the advantages of CNNs and enhanced the accuracy of image semantic segmentation independent of the input image size. The U-Net network [24] based on encoders and decoders, along with the U-Net++ network which improves on the U-Net network, are widely utilized in image segmentation and classification [25]. The composite classification method combined with different CNNs can obtain rich image information to improve vegetation classification performance [26–28]. However, CNNs rely on large amounts of training data, so when the training data is insufficient, the accuracy of vegetation classification will be affected. Urban areas are heavily fragmented, making it challenging to produce sample labels, which also poses a challenge for vegetation classification using CNNs [29]. Additionally, complex scenarios make urban vegetation classification using a single CNN rather tedious.

Therefore, in face of inadequate training labels, we are committed to developing an accurate method for classifying urban vegetation in this paper. The innovative aspects of this paper are as follows: (1) we have explored whether deep-level features that are useful for vegetation classification could be extracted under different FCNs structures; (2) in a highly heterogeneous urban area, our research proposed a joint transfer learning, deep learning, and ensemble learning approach to vegetation classification (UNet++ networks and Stacking, UNS).

## 2. Study Area and Datasets

### 2.1. Study Area

The Chang-Zhu-Tan urban agglomeration (Figure 1), which is located in the middle-eastern part of Hunan Province (28°04′ N, 112°59′ E), includes Changsha City, Zhuzhou City, and Xiangtan City. The study area comprises a humid subtropical monsoon climate with abundant rainfall and rich vegetation types. Over the past four decades, urbanization in Chang-Zhu-Tan has accelerated, making the land types in the urban agglomeration relatively scattered. Based on the Classification of Current Land Use (GB/T 21010-2017), and taking the specific status of vegetation in Chang-Zhu-Tan into account, the main vegetation types to be extracted include grassland, farmland, and forest. Other land types to be extracted include water bodies and built-up land.

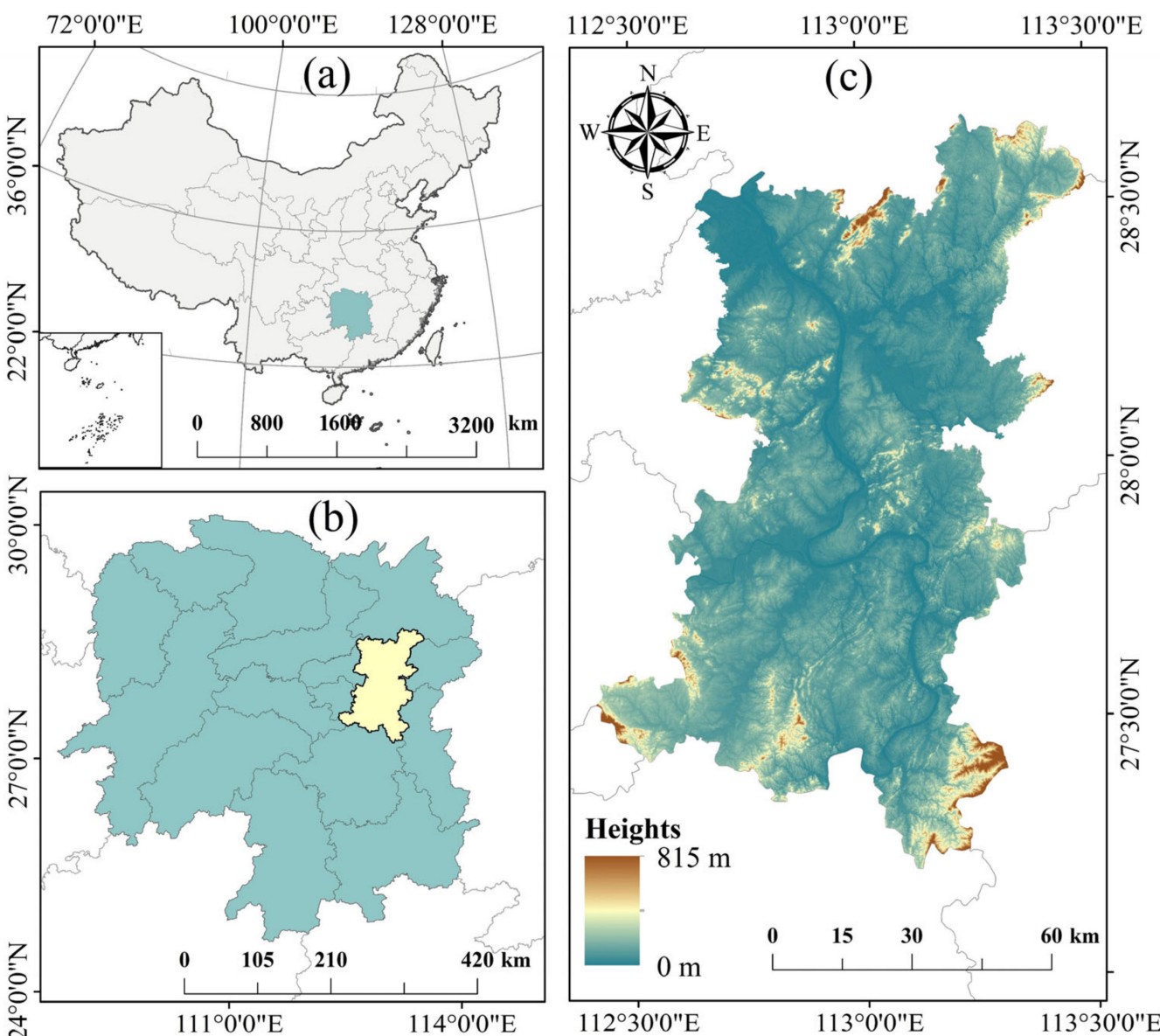

**Figure 1.** Study area: (**a**) the green area is Hunan, China; (**b**) the yellow area is the Chang-Zhu-Tan urban agglomeration; (**c**) DEM data (ASTER GDEM V3 data) of Chang-Zhu-Tan urban agglomeration.

### 2.2. Sentinel-2 Data

The image used in this study was obtained from 2020 Sentinel-2A TOC data (T49RFL, T49RFM, T49RGL, and T49RGM), which was sourced from the Google Earth Engine (GEE).

The Sentinel-2 satellite captured images with 13 bands and varying resolutions of 10 m, 20 m, and 60 m, respectively. Ten bands of Sentinel-2 images at 10 m and 20 m resolution were selected, including visible, near-infrared, and shortwave infrared. The selected images were mostly taken during April and May, when cloud cover and rainfall were infrequent, and image quality was high. After cloud filtering of the Sentinel-2 images with the filter function and de-clouding using the QA60 band on the GEE, the remaining images (10 images) were resampled (nearest neighbor method) to 10 m resolution, median composited, and cropped for further processing.

### 2.3. Other Ancillary Data

Other ancillary data included the administrative division SHP vector file, the DEM data of Chang-Zhu-Tan, and the LULC dataset—i.e., Gaofen Image Dataset (GID) [30]. The China Administrative Divisions SHP vector files were downloaded from the National Basic Geographic Information Centre (http://www.ngcc.cn/ngcc/, accessed on 3 August 2023). DEM data of Chang-Zhu-Tan was available for download in the GEE cloud platform. The GID-5 contained high-quality GF-2 images from over 60 cities in China, which consisted of built-up land, farmland, forest, meadow, water, and unmarked areas. Three full FCNs were pre-trained using the large-scale land covering set (GID-5) from GID to reduce the discrepancy between the migration source and target domains in transfer learning.

### 2.4. Production of Sample Sets

The training sample set has a significant impact on the training results of the deep learning network. However, producing pixel-level labels for urban vegetation classification is difficult due to the fragmentation of land types in the 10 m resolution Sentinel-2 images of the study area. To begin with, sample points were collected from Google Earth images and ground survey data of the same year (Figure 2), the Sentinel-2 images (spectral bands and vegetation indices) covering the study area were classified (89.15%) based on this sample point by the RF algorithm, and provisional labels were obtained. The spectral bands included visible, near-infrared, and shortwave infrared, with a total of 10 bands. The vegetation indices included NDVI, B6RedNDVI, B5TCARI, and B6TCARI, with the following equations:

$$NDVI = (B8 - B4)/(B8 + B4) \tag{1}$$

$$B6RedNDVI = (B8 - B6)/(B8 + B6) \tag{2}$$

$$B5TCARI = 3 \times [(B5 - B4) - 0.2 \times (B5 - B3) \times (B5/B4)] \tag{3}$$

$$B6TCARI = 3 \times [(B6 - B4) - 0.2 \times (B6 - B3) \times (B6/B4)], \tag{4}$$

where B3 is green band; B4 is red band; B5 and B6 are both red edge bands; and B8 is near infrared band. Second, higher-resolution Google Earth imagery combined with ground survey data and data from other periods (Sentinel-2's July, August, and September imagery) was used to manually correct and annotate the provisional labels and modify the misclassifications of land cover types in the labels. Finally, we validated the modified labels using sample points collected from Google Earth imagery and ground survey data from the same year in order to objectively determine the accuracy of the training sample.

After obtaining the final labels, we cropped the labels and the Sentinel-2 image according to a size of 256 × 256, respectively, in Python to obtain 350 sample images and labels. The data augmentation was performed based on the sample images and labels for the purpose of improving the generalization ability of FCNs and the final training efficiency. The sample images and labels were enhanced to four times the original using flipping, mirroring, etc., to compose the final sample set. Then, 70% of the sample set was randomly selected for training, 20% for validation, and 10% for testing.

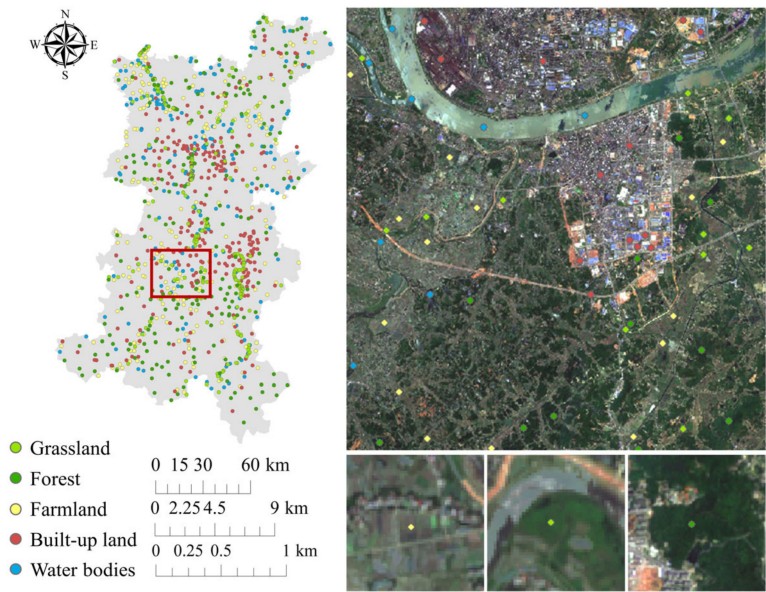

**Figure 2.** Sample points needed to produce labels.

## 3. Methods

Section 3 of this paper focuses on describing methods for the classification of urban vegetation (Figure 3). In this study, we first pre-trained the UNet++, VGG16-UNet++, and ResNet50-UNet++ by transfer learning before extracting the features in the case of insufficient samples. Subsequently, the Relief-F algorithm was used to rank the importance of the extracted deep features, and the optimal feature sets were formed by filtering the top-ranked features to prevent data redundancy. Finally, the optimal feature sets were put into the ensemble learning training program to obtain the final urban vegetation classification results.

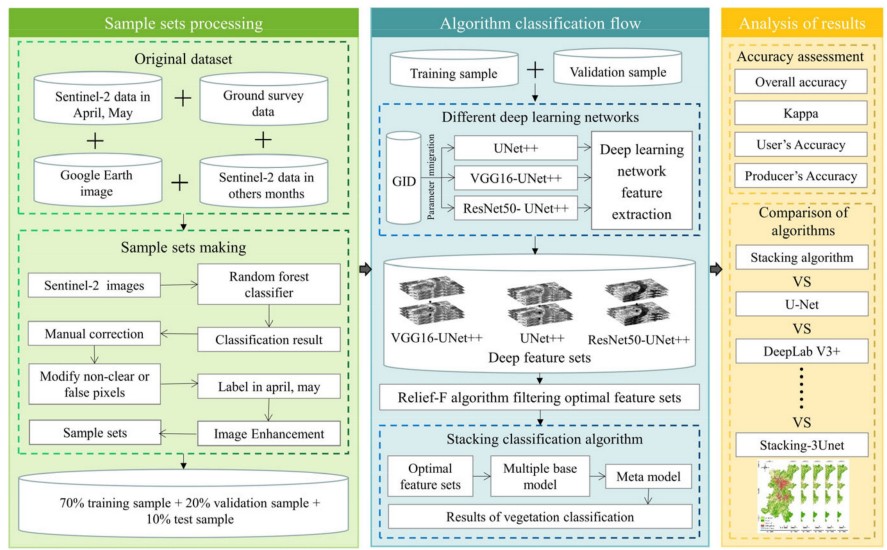

**Figure 3.** Work flow of this study.

### 3.1. Feature Extraction Networks

In the phase of extracting features from images by FCNs, this study used the Tensor-Flow deep learning framework and Python to build three UNet++ networks with different skeletons. The specific versions of TensorFlow and Python are 2.6.0 and 3.6.6, and the graphics card used for the environment was a NVIDIA GeForce GTX 1660. The three different skeletons of UNet++ networks were the normal UNet++ network and the UNet++

network with down-sampling replaced by VGG16 or ResNet50. Transfer learning was performed prior to the training of these networks. The process of transfer learning was intended to pre-train the network on the source domain (GID) to obtain the parameters, and then, when training the network with the produced samples, only to fine-tune the parameters so that the model could be adapted to the present dataset, thus saving a lot of training time. When training the network after pre-training, after several studies to debug the parameters in order to make the CNNs iterate faster during the training process, the parameters were set as follows: the number of iterations was 120; the computational batch size was 4; the optimizer was Adam; and the learning rate was 0.0001. Moreover, if the validation loss rate still did not decrease after 3 epochs, the current learning rate was halved.

### 3.1.1. UNet++

UNet++ is an extended network of the U-Net network which combines the principles of DenseNet [31] and deep supervision [32] on the basis of U-Net, making improvements via three aspects: jumping paths; dense jumping connections; and deep supervision. To first bridge the semantic differences between the down-sampled and up-sampled feature mappings, UNet++ added several convolutional layers to the jump-connected path of each layer of the U-Net (green circles in Figure 4). The output of each convolutional layer incorporated the corresponding up-sampled output of the lower layer. In addition, the jump paths of UNet++ were dense jump connections (blue line in Figure 4). Dense jump connections ensure that the convolutional layers on each jump path reach the correct node, which increases the classification accuracy and improves the network gradient flow. Moreover, UNet++ added deep supervision in the output. The complete UNet++ was trained to give four outputs (L1, L2, L3, L4), each of which corresponded to a different depth of the U-Net network training results. Therefore, each output may have validity and independence. The study was able to adjust the network complexity and improve the classification efficiency of the model via deep supervised pruning of the model.

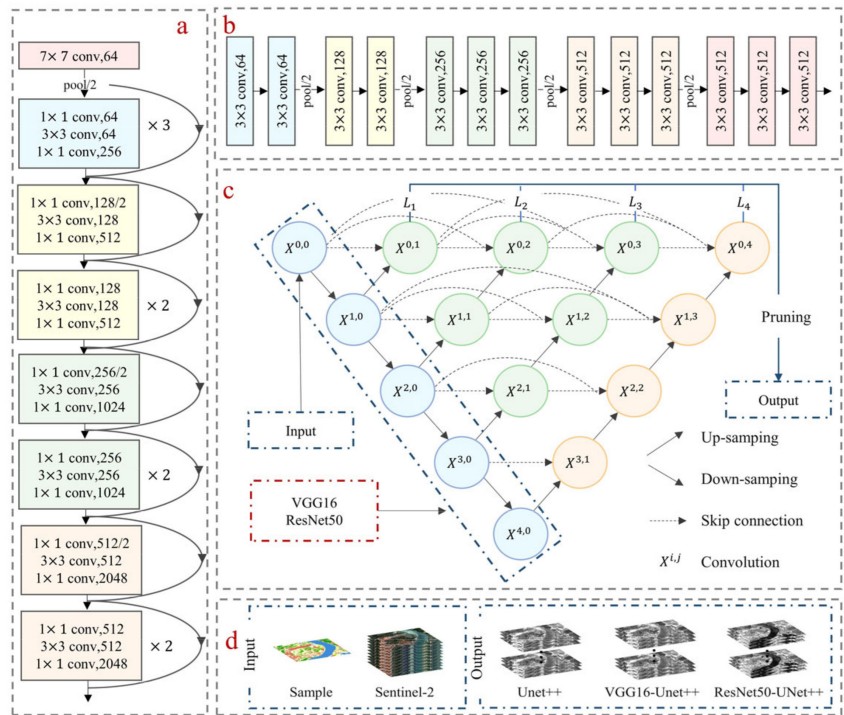

**Figure 4.** Structures of UNet++, VGG16-UNet++, and ResNet50-UNet++ networks: (**a**) structure of ResNet50; (**b**) structure of VGG16; (**c**) structure of UNet++; (**d**) input and output.

The construction of UNet++ featured four down-samples and up-samples, and the number of convolutional kernels in the convolutional layer was set to 32, 64, 128, 256, and 512. Each down-sampling layer contained a maximum pooling layer and two convolutional layers. Each up-sampling layer had an up-sampling layer and two convolution layers. The size of the convolutional kernels in the convolutional layer was 3 × 3, and the activation function was rectified linear unit (ReLu). The size of the convolutional kernels in the pooling layer was 2 × 2. The magnification set in the up-sampling layer was 2. Considering the iteration speed during training, the optimizer chose Adam [33], which converged faster during the training process.

### 3.1.2. VGG16-UNet++

The VGG16-UNet++ network was designed to replace the down-sampled part of the UNet++ network (backbone) with VGG16 (red box in Figure 4). The VGG16 network increased the depth of the network through multiple non-linear layers when controlling the number of network parameters, which improved the training effect of the neural network to a certain extent. The VGG16 network model consisted of 13 convolutional layers and 3 fully connected layers. Only the first 13 convolutional layers of VGG16 were taken for the down-sampling part of UNet++. The size of the convolutional kernel for each layer was 3 × 3, which was filled by the same style, and its structure was shown in Figure 4b.

### 3.1.3. ResNet50-UNet++

The ResNet50-UNet++ network was designed to replace the down-sampled part of the UNet++ network (backbone) with ResNet50 (red box in Figure 4). The ResNet network introduces the residual connected construction of residual cells as a means of solving problems such as gradient disappearance and gradient explosion. The ResNet50 network could be seen as seven parts. This study removed the pooling in the sixth part and the fully connected layer in the seventh part and took only the first five parts of the convolutional layer. The first part contained operations such as convolution, regularization, and activation functions, while the last four parts of the convolution all contained blocks of residuals (Figure 4a).

### 3.1.4. Activation and Loss Functions

The activation function and loss function played an important role in the optimization phase of the network model. The convolution was followed by the addition of an activation function, which enhanced the fit of the network by adding an element of nonlinearity. The advantages of the ReLu function include sparse activation, efficient gradient propagation, and low computational load [34,35]. Therefore, the common ReLu function was chosen as the activation function for the study so as to keep the convergence rate of the model in a steady state. The input features were set to be vectors x. The expression for ReLu was given as follows:

$$f(x) = \max(0, x). \tag{5}$$

In each batch of network training, the loss function would calculate the difference between the predicted and true values (the loss value), and the lower the loss value testified to the better robustness of the network. To avoid the influence of feature area on accuracy and to improve the stability of model training, the binary cross entropy (BCE) [36] and dice loss [37] functions were chosen to form a hybrid loss function (dice BCE loss) [38]. The expressions of BCE, dice loss, and dice BCE loss are as follows:

$$L_{BCE} = -\frac{1}{N} \sum_{i-1}^{N} [y_i \times ln(p_i) + (1 - p_i) \times ln(1 - p_i)] \tag{6}$$

$$L_{Dice} = 1 - \frac{2\sum_{i=1}^{N} (y_i \times p_i)}{\sum_{i=1}^{N} y_i + \sum_{i=1}^{N} p_i} \tag{7}$$



$$L_{DB} = 0.5L_{BCE} + L_{Dice}, \tag{8}$$

where $L_{BCE}$, $L_{Dice}$, and $L_{DB}$ are the loss rates of BCE, dice loss, and dice BCE loss, respectively; n is the total number of sample pixels; $y_i$ is the actual label value of the ith pixel; and $p_i$ is the probability that the ith pixel is predicted to be true [28].

### 3.2. Relief-F Filtering Features

In order to prevent data redundancy and improve classification efficiency, the study extracted features and then selected the Relief-F algorithm for deep feature filtering. Relief-F was proposed on the basis that it can solve multi-classification problems and is a good filtering feature filtering algorithm at present. It transformed the multi-classification problem into a number of binary classification problems, thus bringing samples of the same classification category close together and dispersing samples of different classification types. The Relief-F algorithm had the advantages of high computational efficiency and no requirements for data type and size. The main computational procedures of the Relief-F algorithm are as follows:

$$W_i = W_i - \sum_{j=1}^{k} \frac{diff(A_i, R, H_j)}{mk} + \sum_{C \neq Class(R)} \frac{\frac{p(C)}{1-p(Class(R))} \sum_{j=1}^{k} diff(A_i, R, M_j(C))}{mk}, \tag{9}$$

where $W_i$ represents the weight of the ith feature; diff($A_i$, R, $H_j$) represents the distance between sample R and sample $H_j$ on feature A; p(C) represents the proportion of class C to the total sample tree; $M_j$(C) represents the jth nearest neighbor sample in class C; so diff($A_i$, R, $M_j$(C)) then represents the distance between sample R and sample $M_j$(C) on feature $A_i$.

### 3.3. Ensemble Learning Classification

The optimal features set and label set filtered by the Relief-F algorithm were input to ensemble learning for classification so as to obtain the final vegetation classification results. The ensemble learning algorithm selected for this study was the Stacking algorithm, which will improve the robustness and generalization of urban vegetation classification. The Stacking algorithm is a two-level classifier consisting of multiple base classifiers in the first layer and a meta-classifier in the second layer (Figure 5).

The performance of several base classifiers and meta-classifiers affected the final result of the Stacking algorithm classification. SVM had advantages in the case of small samples, non-linearities, and high-dimensional spaces. K-valued nearest neighbors (k-NN) has a proven theoretical basis and can handle multi-classification problems simply and efficiently. Gradient-boosting decision tree (GBDT) [39] and RF succeed in the areas of boosting and bagging, respectively, and both ensure the effectiveness of classification via different mechanisms. Among them, GBDT was able to handle various types of data flexibly and increase the weight of misclassified samples, while RF had high training efficiency and accuracy and had better resistance to interference and overfitting. Therefore, this study selected three machine learning algorithms of SVM, GBDT, and k-NN with high learning ability as base classifiers, considering the adequacy and diversity of base classifiers. In addition, RF, with better generalization ability, was chosen as the meta-classifier in the second layer. The kernel function in SVM was set to radial basis function (RBF) and cache size was set to 1024. The learning rate in GDBT was 0.05, and the subsample was set to 0.5. The value of k (n neighbors) in k-NN was set to 3. The number of decision trees in the RF was set to 100, and the random state was set to 42.

The Stacking algorithm was calculated as follows: the optimal deep feature set was freely combined and divided into a training set $D_1$ and a test set $D_2$. The 3 base classifiers were used to perform k-fold cross-validation on $D_1$ while the second training set, $D_1'$, was obtained. The meta-classifier was then trained using $D_1'$. Afterwards, $D_2$ was input into the trained base classifier to obtain the second test set $D_2' = \{(y_i, z_{1i}, z_{2i}, \cdots, z_{ni})\}_{i=1}^{k}$, where ©

is the *i*th training sample and n represents the nth base classifier. Finally, $D_2'$ was fed into the trained meta-classifier and the final classification result was output.

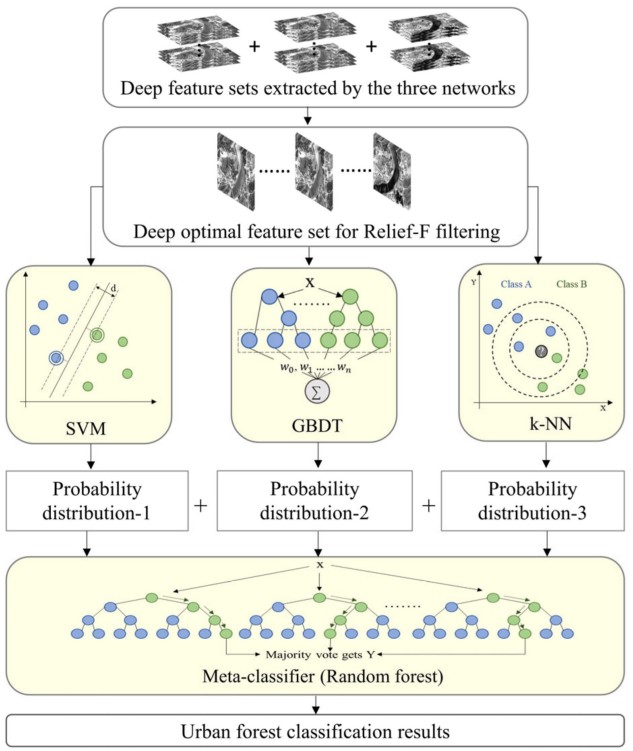

**Figure 5.** Structure of the Stacking algorithm.

*3.4. Accuracy Validation*

Confusion matrices were used in this study to perform accuracy evaluation. The specific evaluation metrics included overall accuracy (OA), Kappa coefficient, producer's accuracy (PA), and user's Accuracy (UA). The different evaluation metrics reflected the accuracy of the classification from different aspects. The overall accuracy and Kappa coefficient were indicators of the overall classification accuracy, while the cartographic accuracy and user accuracy were indicators of missed and wrong classifications for each classification type. The formulae for each accuracy evaluation indicator are as follows:

$$OA = \frac{S_d}{n} \tag{10}$$

$$Kappa = \frac{OA - \frac{\sum (X_{i*} \times X_{*i})}{n^2}}{1 - \frac{\sum (X_{i*} \times X_{*i})}{n^2}} \tag{11}$$

$$PA = \frac{X_{ij}}{X_{i*}} \tag{12}$$

$$UA = \frac{X_{ij}}{X_{*j}}, \tag{13}$$

where $S_d$ is the number of correctly classified samples; n represents the total number of validation samples; $X_{ij}$ represents the number of samples of land cover type j classified as land cover type I; and $X_{i*}$ represents the number of samples classified as land cover type i for the total number of samples. The number of samples with a total of j true land cover types is denoted by the symbol $X_{*j}$.

## 4. Results

*4.1. Deep Features of Network Extraction*

The UNet++, VGG16-UNet++, and ResNet50-UNet++ networks were trained based on the training and validation sets. The flops of UNet++, VGG16-UNet++, and ResNet50-UNet++ networks were 9,040,355, 20,246,819, and 33,203,144, respectively. Four output feature sets (L1, L2, L3, L4) were obtained for each of the three networks, and the loss value variation curves for each output are shown in Figure 6. The loss values of the training and validation sets of the three networks showed relatively consistent trends. The loss curves decreased rapidly in the early training phase and converged gradually in the later phase with increasing training batches, and none of them showed any overfitting phenomenon. VGG16-UNet++ and ResNet50-UNet++ had faster decreasing training and validation loss curves than the UNet++ network and leveled off first at batch 41. The results demonstrated that VGG16 and ResNet50 replacing the down-sampling of the normal UNet++ network would increase the training efficiency of the network. Furthermore, the L4 outputs of the UNet++, VGG16-UNet++, and ResNet50-UNet++ networks had smaller validation loss values than the other three outputs, all stabilizing at around −0.9. Therefore, the L4 outputs of the three networks were the best trained, and the L4 output set was taken for classification in the subsequent classification study.

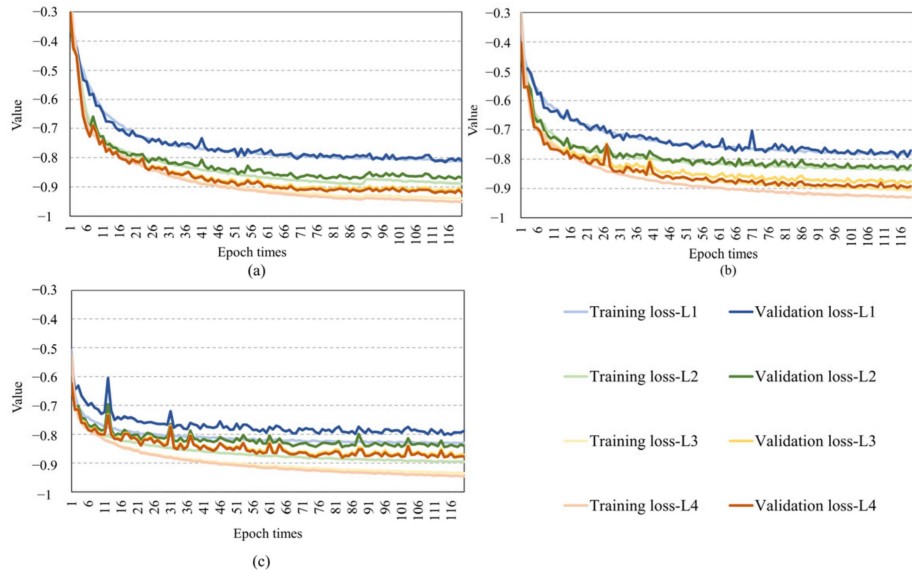

**Figure 6.** Loss curves of different FCNs: (**a**) loss curve of UNet++; (**b**) loss curve of VGG16-UNet++; (**c**) loss curve of ResNet50-UNet++.

In this study, five deep features were selected in the L4 output set of each of the three FCNs, and the differences between different vegetation land types (grassland, forest, and farmland) were analyzed in FCNs via their deep features and the spectral features (Figure 7). The spectral features were reduced by a factor of 100 due to the large difference in the range of the different features. The near-infrared (NIR) and red edge bands of the spectral features were better at separating the three vegetation land types compared to the red, green, and blue bands and the short-wave infrared. However, the NIR and red edge bands were less distinguishable with respect to grassland and forest, while grassland and forest were significantly different in the deep features of UNet++_22, ResNet50-UNet++_16, ResNet50-UNet++_31, and VGG16-UNet++_5, which indicated that the deep features of the three FCNs could better distinguish grassland and forest. In addition, the deep level features in the three FCNs also differed significantly with respect to grassland and forest in terms of information about farmland.

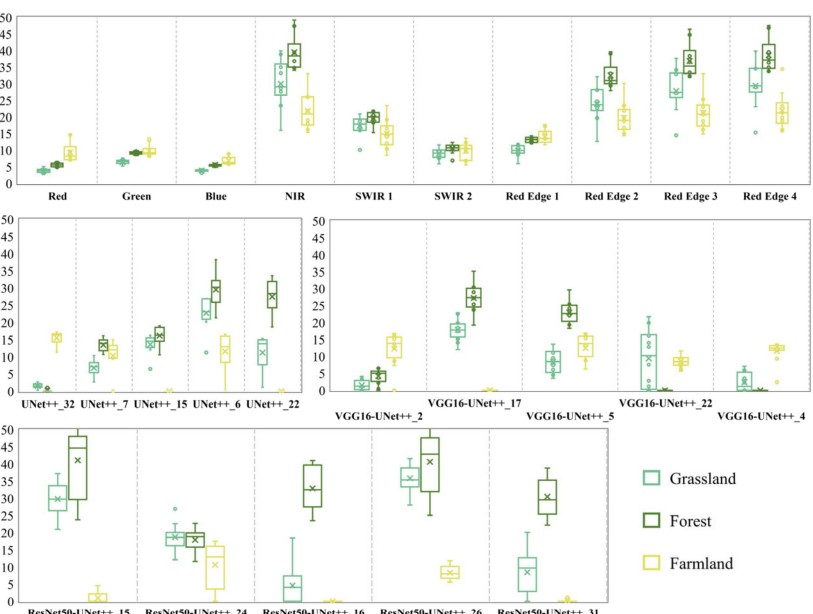

**Figure 7.** The differences for each vegetation land type in the spectral features and in the extraction of deep features via different FCNs (the y-axis is the value of the different features, where the spectral features have been scaled down by a factor of 100).

### 4.2. Results of Deep Features Filtering

Deep features were extracted in this paper by using UNet++ (32), VGG16-UNet++ (32), and ResNet50-UNet++ (32). The deep features (96) extracted by all three networks were filtered based on the Relief-F algorithm, and the feature importance ranking was derived (Figure 8).

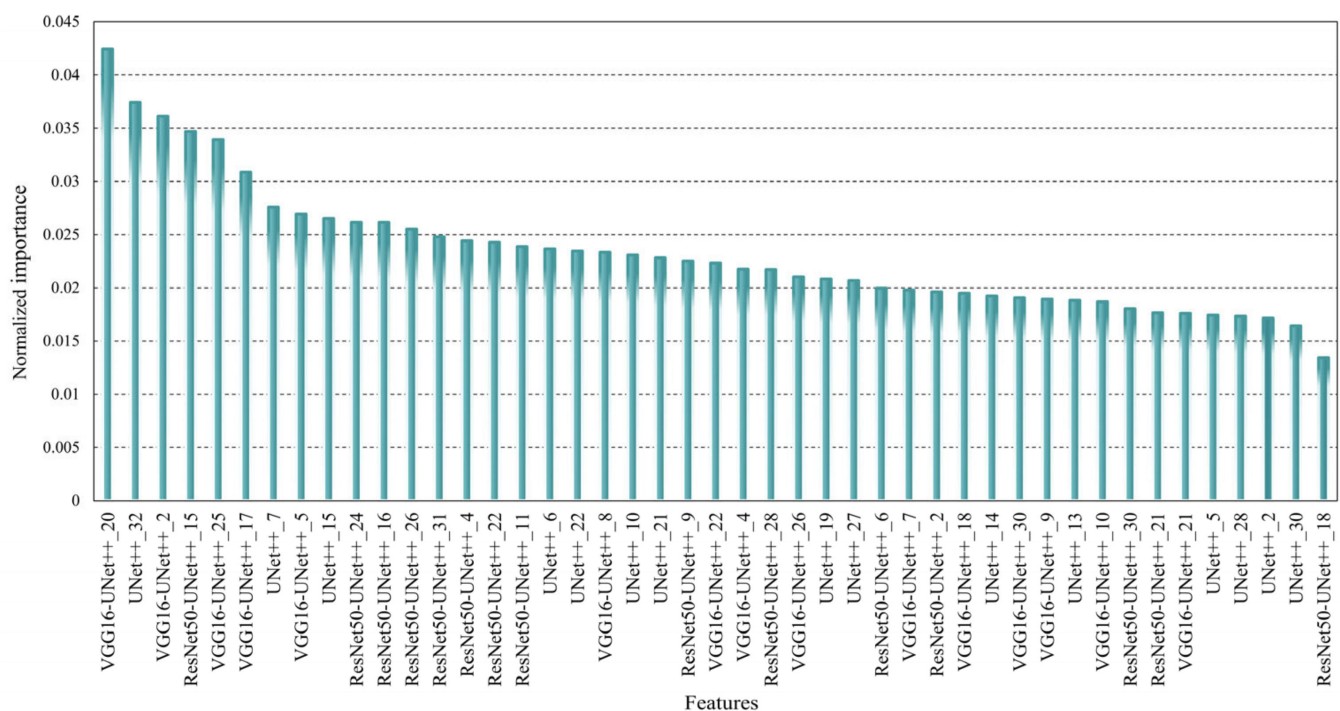

**Figure 8.** Importance ranking of deep features extracted by UNet++, VGG16-UNet++, and ResNet50-UNet++ (top 45 deep features).

UNet++, VGG16-UNet++, and ResNet50-UNet++ networks each had 15 features extracted ranking in the top 45. Among the top 15 features, VGG16-UNet++ extracted features occupied 5, and ResNet50-UNet++ extracted features occupied 7. The number of deep features extracted by VGG16-UNet++ and ResNet50-UNet++ with importance higher than 0.025 was more than the number of deep features extracted by UNet++, which proved that the features extracted by VGG16-UNet++ and ResNet50-UNet++ were superior to those extracted by UNet++ with high semanticity. Considering the computational complexity and efficiency, the top 35 ranked features were selected to form the optimal deep feature set for classification in the subsequent experiments of the study.

### 4.3. Classification Results UNet++ Networks and Stacking Algorithm (UNS)

The urban vegetation classification results by UNS are shown in Figure 9a. A comparison of the classification results via visual discrimination showed that the distribution of the vegetation types (grassland, forest, and farmland) extracted via UNS corresponded to the actual surface conditions. Grassland was scattered throughout the study area, large areas of farmland were mainly located in the north-western part and near the rivers, and forest was evenly distributed in areas other than the cities.

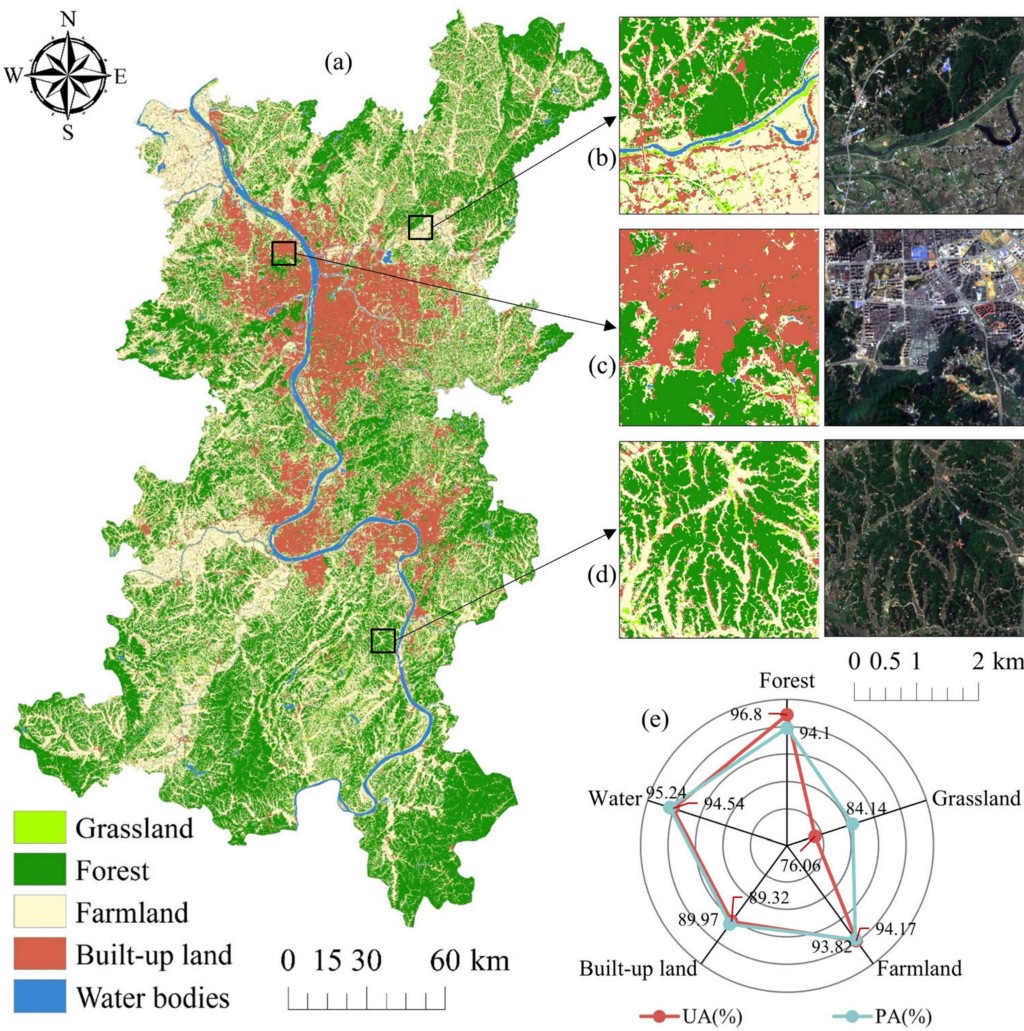

**Figure 9.** Classification results of the UNet++ networks and the Stacking algorithm (UNS): (**a**) classification image of USN; (**b–d**): classification image of three sub-regions; (**e**) classification accuracy of USN.

The OA of the urban vegetation classification results based on UNS was 92.74%, and the Kappa was 0.8905. UA and PA were above 89% for forest and farmland (Figure 9e), but only 76.06% and 84.14% for grassland. The main reasons for this were the small sample of grassland relative to the other land types and the fact that grassland and other vegetation types (forest and farmland) had similar spectral features during parts of the growing season, which influenced the deeper features to some extent.

In addition, the study intercepted a total of three sub-regions from the study area with different heterogeneity, namely, suburban (Figure 9a), urban center (Figure 9b), and mountainous area (Figure 9c), in order to better analyze the classification results of UNS for presentation. Area 1 (Figure 9b) contained regularly shaped farmland, and UNS was able to distinguish well between farmland and built-up land, and between farmland and grassland, and the boundaries of the forest land were clearly and carefully delineated. Area 2 (Figure 9c) was located in an urban agglomeration with a high degree of land fragmentation, and UNS extracted the fragmented forest in its entirety. However, as some of the farmland was not planted with crops or cash crops in April and May, the features of the farmland and the built-up bare soil were similar, making it prone to misclassification. Area 3 (Figure 9d) was located in mountainous terrain, and the method provided accurate mapping of both forest and farmland. In general, UNS enabled the classification of urban vegetation from high-resolution imagery.

## 5. Discussion

### 5.1. Analysis of Transfer Learning

To verify the effectiveness of transfer learning, urban vegetation was classified by UNet++, VGG16-UNet++, and ResNet50-UNet++ without transfer learning (a, b, and c) and with transfer learning (d, e, and f), and accuracy evaluation results (Figure 10) were obtained. The results showed that the transfer learning followed by retraining UNet++, VGG16-UNet++, and ResNet50-UNet++ networks improved OA and Kappa by 4.86% and 0.0843, 5.75%, and 0.0906, and 5.85% and 0.0918, respectively, compared to the networks without transfer training. Of these, ResNet50-UNet++ with transfer learning resulted in the greatest improvement in OA and Kappa. The transfer learning was shown to reduce the generalization error and improve the training efficiency of the network in the case of insufficient samples, which is consistent with previous research findings [40].

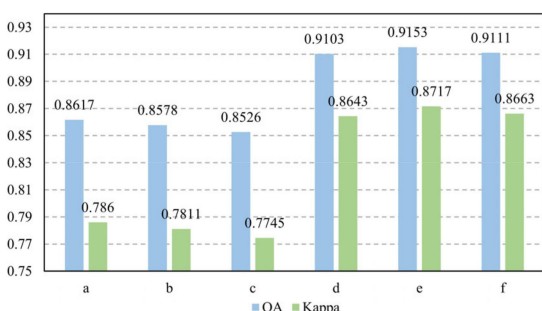

**Figure 10.** Classification results of the networks after transfer learning and the without transfer learning: (**a**) UNet++ without transfer learning; (**b**) VGG16-UNet++ without transfer learning; (**c**) ResNet50-UNet++ without transfer learning; (**d**) UNet++ with transfer learning; (**e**) VGG16-UNet++ with transfer learning; (**f**) ResNet50-UNet++ with transfer learning.

### 5.2. Classification Results by Single FCNs or Stacking

Single FCNs or Stacking were also used to classify the vegetation, and the results were compared with those of UNS. Compared to the single FCNs UNet++, VGG16-UNet++, ResNet50-UNet++, and the Stacking algorithm, the UNS improved the OA by 2.34%, 1.8%, 2.29%, and 10.74%, respectively (Table 1). In terms of the UA and PA for forest and farmland, the UNS were all improved to varying degrees compared to single FCNs or Stacking. For

UA in grassland, the UNS improved by 14.78%, 6.76%, 17.31%, and 3.34% over UNet++, VGG16-UNet++, ResNet50-UNet++ and Stacking, respectively (Figure 9 and Table 1).

**Table 1.** Classification accuracy of single FCNs or Stacking.

| Land Types | Stacking | | UNet++ | | VGG16-UNet++ | | ResNet50-UNet++ | |
|---|---|---|---|---|---|---|---|---|
| | UA (%) | PA (%) | UA (%) | PA (%) | UA (%) | PA (%) | UA (%) | PA (%) |
| Forest | 94.53 | 85.99 | 94.75 | 92.03 | 95.11 | 92.43 | 95.17 | 91.36 |
| Grassland | 72.63 | 52.82 | 61.28 | 81.86 | 69.30 | 77.75 | 58.15 | 87.16 |
| Farmland | 76.53 | 91.60 | 92.52 | 91.80 | 92.75 | 92.55 | 91.17 | 92.69 |
| Built-up land | 82.72 | 72.87 | 87.53 | 82.85 | 86.85 | 87.72 | 91.49 | 82.79 |
| Water bodies | 96.32 | 72.65 | 92.91 | 93.59 | 94.24 | 92.66 | 93.60 | 93.70 |
| OA | 82.02% | | 90.40% | | 90.94% | | 90.45% | |
| Kappa | 0.7409 | | 0.8542 | | 0.8628 | | 0.8560 | |

In addition, the classification OA and Kappa of all three FCNs were above 90% and 0.85. Among them, the VGG16-UNet++ network had the highest urban vegetation classification accuracy with classification OA and Kappa coefficients of 90.94% and 0.8628, respectively (Table 1). From the above, this indicated that the deeper features extracted by the VGG16-UNet++ network were beneficial in improving the classification results compared to the other two FCNs. Su et al. [41] achieved better results in the semantic segmentation of remote sensing images using a model of MFNet with VGG16 as the backbone on the Potsdam test set, which is consistent with the results of this study. In addition, the classification OA and Kappa of the Stacking algorithm were only 82.02% and 0.7409, which were 8.38%, 8.43%, and 8.92% lower than those of UNet++, ResNet50-UNet++, and VGG16-UNet++, respectively. Furthermore, the visual analysis (Figure 11) revealed some obvious misclassifications in the results of the Stacking algorithm compared to the results of the three FCNs. In the middle of the classification map, some built-up pixels were misclassified as farmland by the Stacking algorithm, which did not correspond to the actual situation. This partly indicated that the FCNs were capable of extracting features from remote sensing images and thus of improving the results of vegetation classification compared to the Stacking algorithm.

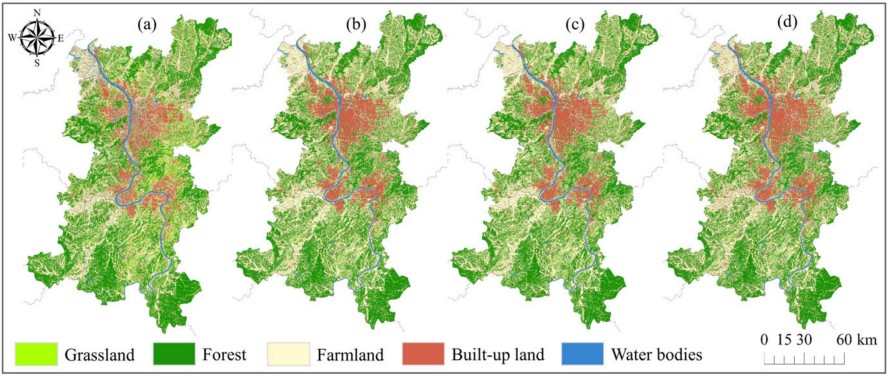

**Figure 11.** Classification results of single FCNs or Stacking: (**a**) classification results of Stacking; (**b**) classification results of UNet++; (**c**) classification results of VGG16-UNet++; (**d**) classification results of ResNet50-UNet++.

### 5.3. Classification Results by Using Different FCNs Combined with Stacking

The convolution layer has a rich feature set to improve classification. Therefore, in order to verify the urban vegetation classification accuracy with different deep features, the deep features extracted by each of UNet++, VGG16-UNet++, and ResNet50-UNet++, as well as the optimal deep feature set obtain after filtering by three types of FCNs extracted

features, were combined with the Stacking algorithm for urban vegetation classification, respectively, and the accuracy evaluation was shown in Figure 12.

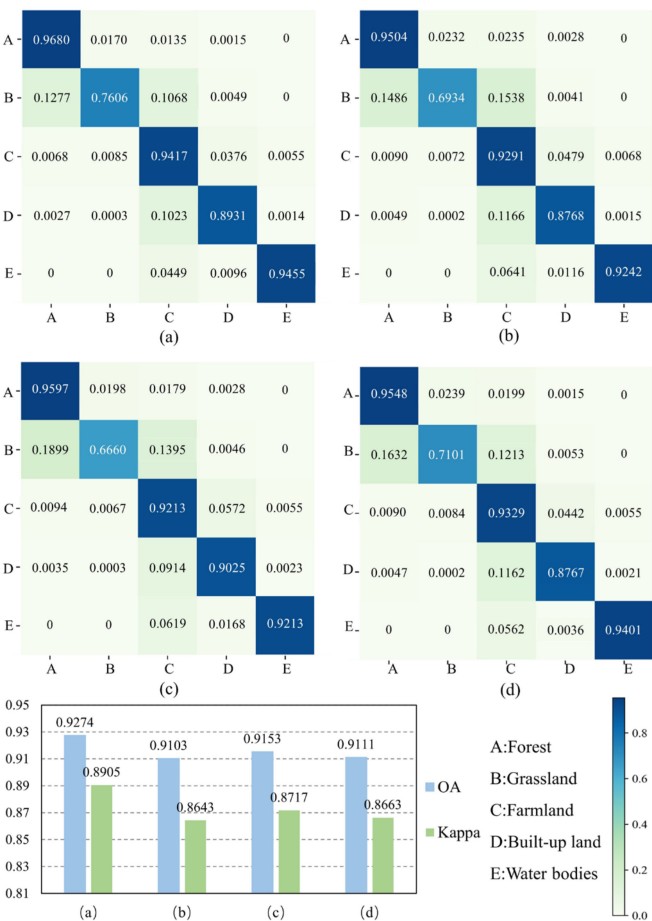

**Figure 12.** Classification accuracy of different FCNs combined with Stacking: (**a**) classification accuracy of UNS; (**b**) classification accuracy of UNet++ combined with Stacking; (**c**) classification accuracy of VGG16-UNet++ combined with Stacking; (**d**) classification accuracy of ResNet50-UNet++ combined with Stacking.

The single FCNs and the Stacking algorithm combined (b, c, and d) all had accuracies above 91%, which proved that the features trained and extracted using the FCNs had good separability. Among them, VGG16-UNet++ extracted deep features better than UNet++ and ResNet50-UNet++, which achieved UA and Kappa of 91.53% and 0.8717. UNS (a) improved the OA by 1.71%, 1.21%, and 1.63%, respectively, and the kappa coefficients by 0.0262, 0.0188, and 0.0242, respectively, compared with a single FCNs combined with the Stacking algorithm (b, c, and d). From the confusion matrix of the four methods, the deep features extracted by methods (b), (c), and (d) performed differently for different land types. (d) had a higher UA of forest compared to (b) and (c), and (c) had a higher UA of forest and farmland compared to (c) and (d). However, the UA of vegetation (grassland, forest, and farmland) obtained via the method combining deep features extracted by all three FCNs was improved compared to the other three methods. The highest UA improvement was obtained for grassland, with (a) being 6.74%, 9.46%, and 5.05% higher than the UA for (b), (c), and (d), respectively. Therefore, image features extracted via a single model are often inadequate compared to those extracted by multiple models [42], and the study was able to effectively improve the classification accuracy by fusing features with differences.

### 5.4. Classification Results by Other Vegetation Classification Methods

It has previously been shown that U-Net and DeepLab V3+ have achieved some results in vegetation classification [43]. Therefore, to further validate the effectiveness of UNS, RF, U-Net, and DeepLab V3+ were used to extract urban vegetation in this paper, and the classification results were compared with those of UNS. The results showed (Table 2) that the OA of RF, U-Net, and DeepLab V3+ were all above 81%, with the OA (89.69%) and Kappa coefficient (0.8432) of U-Net being higher than those of RF and DeepLab V3+. Compared to RF, U-Net, and DeepLab V3+, UNS improved in OA by 11.31%, 9.38%, and 3.05%, and in both vegetation UA and PA, respectively. The largest increases were shown in grassland UA and PA, with UNS improving grassland UA and PA by 10.4% and 36.78%, 17.31% and 8.43%, and 12.65% and 8.8% over RF, U-Net, and DeepLab V3+, respectively.

**Table 2.** Classification accuracy of different image vegetation classification methods: (**a**) classification accuracy of UNet++ networks and Stacking; (**b**) classification accuracy of U-Net; (**c**) classification accuracy of DeepLab V3+; (**d**) classification accuracy of RF.

| Land Types | (a) | | (b) | | (c) | | (d) | |
|---|---|---|---|---|---|---|---|---|
| | UA (%) | PA (%) | UA (%) | PA (%) | UA (%) | PA (%) | UA (%) | PA (%) |
| Forest | 96.80 | 94.10 | 96.73 | 88.35 | 93.74 | 78.09 | 93.89 | 85.18 |
| Grassland | 76.06 | 84.14 | 63.41 | 75.34 | 58.74 | 75.71 | 65.66 | 47.36 |
| Farmland | 94.17 | 93.82 | 90.12 | 93.11 | 81.80 | 89.32 | 76.36 | 91.36 |
| Built-up land | 89.32 | 89.97 | 87.26 | 84.87 | 82.77 | 74.47 | 82.53 | 73.13 |
| Water bodies | 94.54 | 95.24 | 94.68 | 91.62 | 88.26 | 92.15 | 96.32 | 72.65 |
| OA | 92.74% | | 89.69% | | 83.36% | | 81.43% | |
| Kappa | 0.8905 | | 0.8452 | | 0.7531 | | 0.7326 | |

Four subareas were selected from the study area—farmland, suburban, urban, and mountainous areas—to qualitatively analyze the classification results of UNS. First, the ungrown farmland was easily misclassified into built-up land and water bodies due to the close proximity of the ungrown farmland and built-up bare land features. Compared to the other methods that used convolution to extract features, RF clearly misclassified the farmland in region (a) into built-up land and water bodies. In the three methods that used convolution (UNS, U-Net, and DeepLab V3+), UNS reduced the built-up land misclassification into farmland compared to the other networks (Figure 13C). Furthermore, UNS and RF distinguished grassland more accurately than U-Net and DeepLab V3+ in region (b). UNS reduced the loss of forest boundaries compared to the results derived by RF. Forest was misclassified as grassland in the RF and DeepLab V3+ results (Figure 13C,D) due to the effect of mountain shading. In contrast to U-Net, UNS showed less "salt-and-pepper" (Figure 13D). Therefore, combining deep learning and ensemble learning, UNS was able to enhance the generalization and robustness of the model by combining deep features extracted by different networks and effectively extracting information on urban vegetation over large areas [44].

### 5.5. Limitations and Future Work

Dong et al., combined the techniques of FCN and RF for the classification of high-resolution remote sensing images and achieved excellent classification results, but only a single FCN was utilized, and features may be insufficient [26]. In contrast, this study utilized three FCNs combined with ensemble learning for urban vegetation classification and achieved good classification results. However, as only single-time phase Sentinel-2 images were used in this study, resulting in poor classification of the grassland type, subsequent vegetation classification could be carried out using multi-temporal data. In addition, the parameters and loss functions of the model could be further refined and subsequently adjusted to improve the classification accuracy. Alternatively, a network with better

spatial generalization capability can be used instead of FCNs to address spatiotemporal heterogeneity [45].

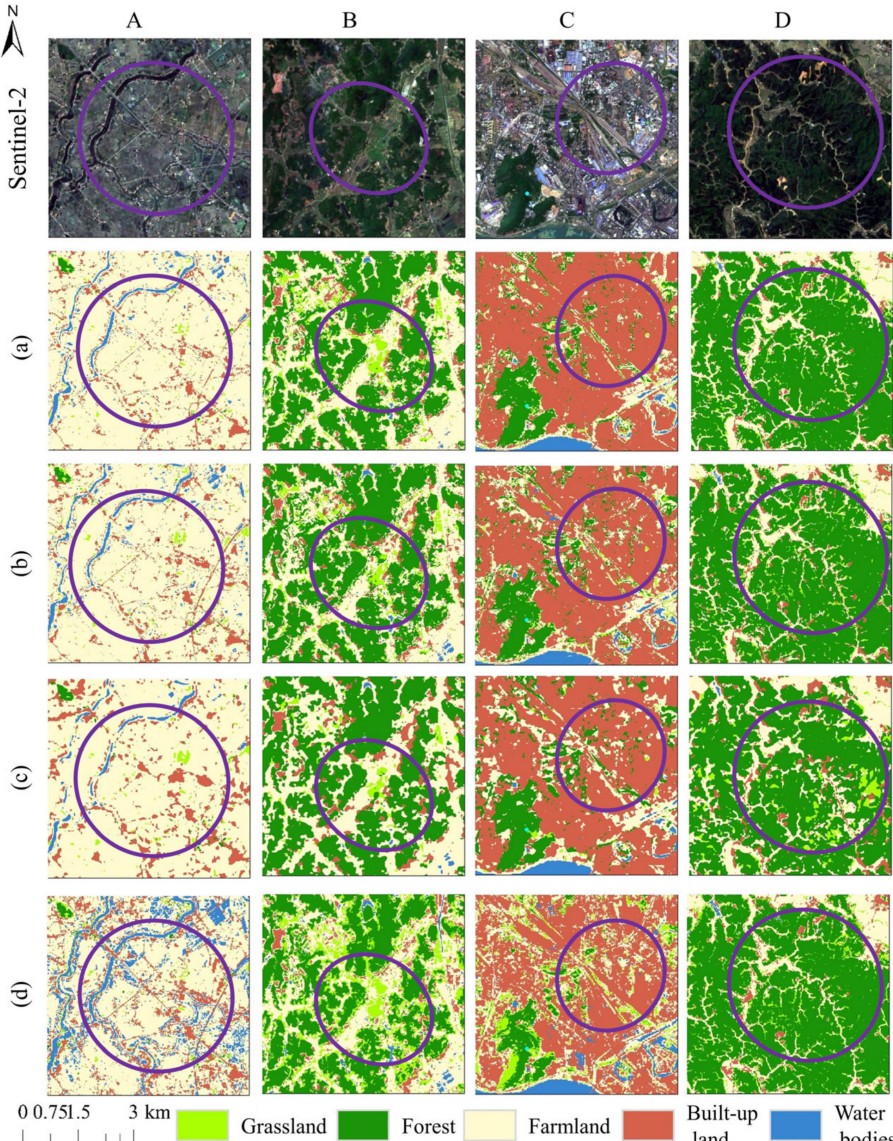

**Figure 13.** Results of vegetation classification with RF, U-Net, and DeepLab V3+ for four typical areas (**A**–**D**): (**a**) classification results of UNet++ networks and Stacking; (**b**) classification results of U-Net; (**c**) classification results of DeepLab V3+; (**d**) classification results of RF.

## 6. Conclusions

To address the challenge of accurately identifying vegetation in urban areas, this study proposed a vegetation classification method (UNS) that combines the advantages of transfer learning, FCNs, and ensemble learning. The results indicated that the deep features extracted by UNet++, VGG16-UNet++, and ResNet50-UNet++ were more effective than the spectral features alone for identifying vegetation. The fusion of the deep features extracted by UNet++, VGG16-UNet++, and ResNet50-UNet++ improved the classification of urban vegetation. In addition, the research method effectively displayed the vegetation information of the images and provided better classification of urban vegetation than common vegetation classification algorithms. In conclusion, the study provides a methodological reference for the extraction of urban vegetation information in the face of accelerating urbanization.

**Author Contributions:** Conceptualization, S.C., M.Z. and F.L.; Data curation, S.C. and M.Z.; Funding acquisition, M.Z.; Methodology, S.C. and M.Z.; Software, S.C. and M.Z.; Validation, M.Z. and F.L.; Writing—original draft, S.C. and M.Z.; Writing—review and editing, M.Z. All authors have read and agreed to the published version of the manuscript.

**Funding:** This study was funded by the National Natural Science Foundation of China (41901385), the Natural Science Foundation of Hunan Province of China (2022JJ40873), the Education Department of Hunan Province of China (21A0177), and, in part, by the Key Laboratory of Natural Resources Monitoring and Supervision in the Southern Hilly Region, Ministry of Natural Resources (NRMSSHR-2022-Y06).

**Data Availability Statement:** Since the sample points dataset are the part of the author' graduation paper, they are not publicly available at the moment. The other data can be obtained from the means provided in the text.

**Acknowledgments:** The urban vegetation code method of this study is openly available at https://github.com/Shu-Dan/vegetation-classification/ accessed on 3 August 2023. The authors would like to thank the anonymous reviewers and members of the editorial team for their comments and contributions.

**Conflicts of Interest:** The authors declare no conflict of interest.

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
