# Peer review of "Mapping Vegetation Types by Different Fully Convolutional Neural Network Structures with Inadequate Training Labels in Complex Landscape Urban Areas"

_forests, doi:10.3390/f14091788_

Round 1

Reviewer 1 Report

Review report for "Mapping vegetation types by different fully convolutional neural network structures with inadequate training labels in complex landscape urban areas"

 The paper is methodologically well done, followed by appropriate literature. The topic is current. The paper integrates methodological solutions which already exist, which is my great concern and remark.

The paper utilized three U-Net to classify the complex landscape by implementing ensemble and transfer learning process.

Some points needs clarifications

1. As U-Net works better than VGG and ResNet was a known fact. So what motivates the author to compare the classification acurracy of the utilized three models.

2. The related work was carried out but the principle of the proposed method should be very clear.

3. The advantages and disadvantages of the previous work are not clearly expounded, in other words, the motivation of writing the paper is not explained.  The following paper can be cited to improve the quality of the paper

https://doi.org/10.3390/app13053210

https://doi.org/10.3390/rs15010006

4. As UNet takes much time for training and testing, the authors can clearly mention the working environment of the proposed work.

5. The proposed model is based on UNet with ensemble model. Why the comparative study was carried out along with RF as mentioned in Table 2.

6. The computational complexity of the proposal should be included.

7. What are the hyper parameters tuned to identify the converging points of the three models.

8.Figure 6,7,and 8 should be redrawn as it is in a readable format.

9. It would be nice to make a picture of the algorithm of the realized research.

Moderate English  grammar and Spell check required.

Author Response

Thanks very much for these comments and please find our detailed response to each of your suggestions in the attachment. 

Reviewer 2 Report

Many thanks to the Authors, as they presented well-designed and clear-structured study.

Minor criticism can be proposed:

- the “%” symbol have to be split from previously written number by white space – white spaces are missed in all the text;

- similarly, white spaces are missed before references (“[]” symbols) in Introduction section, and in some places bellow (line 283, for instance);

- similarly, white spaces are missed in line 48 (“/advanced”), line 112 (“10m and 20m”), line 132 (“10m”), line 222 (“ResNet50(”);

-it is better to cut the paragraph in page 17 after third sentence (in line 496), and move all the sentences of the paragraph beginning from this line (lines 496-506) to insert after Figure 13 – to exclude cut of the Figure caption by page break;

- Figure 1 – scale bars are needed in all three maps, while it is better to use single line scale bar (same to the used in Google Maps) instead of “taxi” style scale bar used currently;

- Figure 1c – “M” in the legend have to be replaced to “m”, “Km” have to be replaced to “km”, “DEM” have to be replaced to “Heights”;

- Figure 2 – single line scale bars have to be added in the map and in all four satellite images;

- Figure 9 – scale bars have to be added in map/satellite fragments in upper right side, already presented scale bar is better to be replaced into single line scale bar, “Km” have to be replaced to “km”;

- Figure 11 – scale bar is better to be replaced into single line scale bar, “Km” have to be replaced to “km”;

- Figure 13 – scale bars have to be added in satellite fragments in the first row.

Author Response

(The authors gave the same response as above.)

Round 2

Reviewer 1 Report

The authors have given their responses in a satisfactory manner. The manuscript may be accepted for publication. 

Moderate English grammar and spell check is required